# HEART rate variability biofeedback for long COVID symptoms (HEARTLOC): protocol for a feasibility study

Joanna Corrado,[1,2] Stephen Halpin [ORCID] ,[1,3,4] Nick Preston [ORCID] ,[1] Diana Whiteside,[3] Rachel Tarrant,[3] Jennifer Davison,[3] Alexander D Simms,[5] Rory J O'Connor [ORCID] ,[1,2] Alexander Casson [ORCID] ,[1,4] Manoj Sivan [1,2,3,4]

[1]Academic Department of Rehabilitation Medicine, University of Leeds, Leeds, UK
[2]National Demonstration Centre in Rehabilitation, Leeds Teaching Hospitals NHS Trust, Leeds, UK
[3]Long COVID Rehabilitation Service, Leeds Community Healthcare NHS Trust, Leeds, UK
[4]The University of Manchester, Manchester, UK
[5]Department of Cardiology, Leeds Teaching Hospitals NHS Trust, Leeds, UK

**Correspondence to**
Dr Manoj Sivan;
m.sivan@leeds.ac.uk

## ABSTRACT

**Introduction** Long COVID (LC), also known as post-COVID-19 syndrome, refers to symptoms persisting 12 weeks after COVID-19 infection. It affects up to one in seven people contracting the illness and causes a wide range of symptoms, including fatigue, breathlessness, palpitations, dizziness, pain and brain fog. Many of these symptoms can be linked to dysautonomia or dysregulation of the autonomic nervous system after SARS-CoV2 infection. This study aims to test the feasibility and estimate the efficacy, of the heart rate variability biofeedback (HRV-B) technique via a standardised slow diaphragmatic breathing programme in individuals with LC.

**Methods and analysis** 30 adult LC patients with symptoms of palpitations or dizziness and an abnormal NASA Lean Test will be selected from a specialist Long COVID rehabilitation service. They will undergo a 4-week HRV-B intervention using a Polar chest strap device linked to the Elite HRV phone application while undertaking the breathing exercise technique for two 10 min periods everyday for at least 5 days a week. Quantitative data will be gathered during the study period using: HRV data from the chest strap and wrist-worn Fitbit, the modified COVID-19 Yorkshire Rehabilitation Scale, Composite Autonomic Symptom Score, WHO Disability Assessment Schedule and EQ-5D-5L health-related quality of life measures. Qualitative feedback on user experience and feasibility of using the technology in a home setting will also be gathered. Standard statistical tests for correlation and significant difference will be used to analyse the quantitate data.

**Ethics and dissemination** The study has received ethical approval from Health Research Authority (HRA) Leicester South Research Ethics Committee (21/EM/0271). Dissemination plans include academic and lay publications.

**Trial registration number** NCT05228665.

---

## STRENGTHS AND LIMITATIONS OF THIS STUDY

⇒ To our knowledge, this is the first study of heart rate variability biofeedback (HRV-B) in long COVID and will provide new information regarding the feasibility of the technology-based intervention in this condition.

⇒ The estimation of efficacy will determine the scope and sample size for a larger controlled trial in the condition that currently has no definitive treatments.

⇒ The study will provide preliminary evidence on the correlation between long COVID symptoms and dysautonomia.

⇒ The limitation of this study is the small sample size of 30 participants, which might not give an accurate estimate of efficacy.

⇒ HRV-B is a technology-based intervention; therefore, its take-up could be limited in those with a lack of experience in using digital technology in daily life, particularly those from less privileged backgrounds.

## INTRODUCTION

Post-COVID-19 syndrome or Long COVID (LC) refers to persistent symptoms 12 weeks after SARS-COV2 infection and can include symptoms of physical fatigue, cognitive fatigue or 'brain fog', breathlessness, pain and psychological distress.[1–3] An estimated 1.4 million people are reported to be affected by LC in the UK alone.[4] The condition can be highly debilitating for some, particularly middle-aged individuals who were previously functioning at a high level and in demanding vocational roles.[5] Many will experience significant disruption to employment, social and caregiving roles and participation in society.

Many LC symptoms such as palpitations, dizziness, fatigue, pain and breathlessness can be explained by the theory of dysautonomia.[6 7] This is a state of episodic dysregulation in the autonomic nervous system (ANS) with sympathetic overdrive and reduced parasympathetic activity. Dysautonomia plays a significant role in the symptomology of many long-term conditions including multiple sclerosis, Parkinson's disease, diabetes mellitus, fibromyalgia, chronic fatigue syndrome and migraine.[8]

One way of estimating and measuring autonomic function is through heart rate variability (HRV), as cardiac rate and rhythm are controlled largely by the ANS. The parasympathetic nervous system chiefly activates

a slowing of heart rate through the vagus nerve, and the sympathetic response acts through the activation of β-adrenergic receptors.[9] HRV can be measured either in the time domain or frequency domain. HRV represents a measure of the variation in time between heartbeats (captured on an Electrocardiogram (ECG) strip as a time interval between the R waves of the QRS complexes). A low HRV is associated with sympathetic nervous system activation, also described as a state of 'fight or flight'. Higher HRV correspond with parasympathetic nervous system activation and is believed to reflect a state of rest and recovery. Lower HRV has been observed to be associated with fatigue and pain symptoms of chronic fatigue syndrome/myalgic encephalomyelitis and fibromyalgia[10–12] as well as other chronic physical and mental health pathologies including asthma, anxiety and stress.[10–14]

## HRV biofeedback

When physiological parameters such as HRV are monitored in real time with self-regulation techniques such as breathing exercises applied to influence the parameters, this is known as biofeedback.[15 16] In this study, for monitoring and modulating the HRV, we are using breathing techniques to encourage the predominance of parasympathetic nervous activity through vagus nerve activation. To the best of our knowledge, there have not yet been any studies of HRV biofeedback (HRV-B) in LC. However, HRV-B using breathing techniques has been tested in other clinical conditions such as asthma,[13] depression[17] and fibromyalgia.[12] A normal respiratory rate is between 12 bpm and 20 bpm.[18] The optimal breathing frequency to produce maximal increase in HRV varies for each individual but on average is between 5.5 bpm and 6 bpm and is known as resonant breathing.[13 18 19] Resonant breathing helps to restore autonomic balance due to baroreflex gain and vagal activation.[13 18–20]

There are several means of assessing HRV but most commonly these include the use of either wearable devices such as smartwatches or chest straps, or through small attachable Holter ECG units. These are non-invasive and readily available, although reliability differs between devices and platforms. Many commercial HRV devices are associated with smartphone app technology, which can be readily downloaded and made available to participants for monitoring. Of the consumer grade devices available to monitor HRV the Polar H10 chest strap is felt to be the most reliable and remains accurate even during high-intensity activity.[21] The Polar H10 can be linked with the Elite HRV app, which provides real-time feedback on HRV and the user's response to breathing techniques. The combination of Polar H10 chest strap and Elite HRV app has been effectively used to harness real-time physiological data, for example, in athletes.[22] In contrast, many wrist-worn devices such as Fitbit return a measure of HRV only while the user is asleep due to motion and other interference sources, meaning real-time HRV-B is not possible.

The aim of this study is to determine the feasibility and impact of a structured HRV-B regime incorporating diaphragmatic breathing exercise, on LC symptoms. We wish to test the acceptability and compliance of the intervention and estimate effect on symptoms using standardised validated measures of LC and dysautonomia.

## Aims and objectives

The aim of this study is: to assess the feasibility of a 4-week HRV-B-structured breathing programme in individuals with LC.

The objectives include:
1. Does breathing exercises through HRV-B increase HRV among participants with LC?
2. Are consumer-grade monitors appropriate technology to use for HRV-B in the domiciliary setting?
3. Does regular HRV-B have any effect on LC symptoms?

## METHODS
### Study design

This is a phase 2 uncontrolled open-label feasibility study of a home technology-based HRV-B in 30 individuals with LC. Potential participants will be identified through the Leeds COVID-19 Rehabilitation Service, based at Leeds Community Healthcare NHS Trust. The study period will be 6 weeks for each participant. The study start date is 24 January 2022, and the anticipated end date is 31 March 2024.

### Eligibility criteria

The inclusion criteria are
► Age ≥18 years.
► Confirmed LC diagnosis as per the National Institute for Health and Care Excellence (NICE) criteria for post-COVID syndrome.[1]
► Self-rating of at least 'moderate' or 'severe' on dysautonomia questions of palpitations or dizziness on the COVID-19 Yorkshire Rehabilitation Scale (C19-YRSm).[23]
► Abnormal NASA Lean Test (NLT).[24–26]
   HR increase in 30 bpm or ≥120 bpm.
   Blood pressure (BP) decrease in 20 mm Hg systolic or 10 mm Hg diastolic in the first 3 min of standing.

NLT is an accepted measure of cardiovascular instability and is conducted at initial assessment clinic for all LC service users in the Leeds COVID-19 rehabilitation service. The patient lies down for 2–5 min prior to the test with HR and BP taken each minute to calculate average supine values. They then stand with heels 6 inches from a wall and lean back against it with HR and BP taken each minute for 10 min. Abnormal results (as described above) are demonstrated through orthostatic hypotension or tachycardia on standing which are hallmarks of dysautonomia and, therefore, objectively quantifiable. The participants who have dysautonomia symptoms but do not meet the mentioned thresholds will not be included in this feasibility study but will be potential recruits for future larger scale studies using the same intervention.

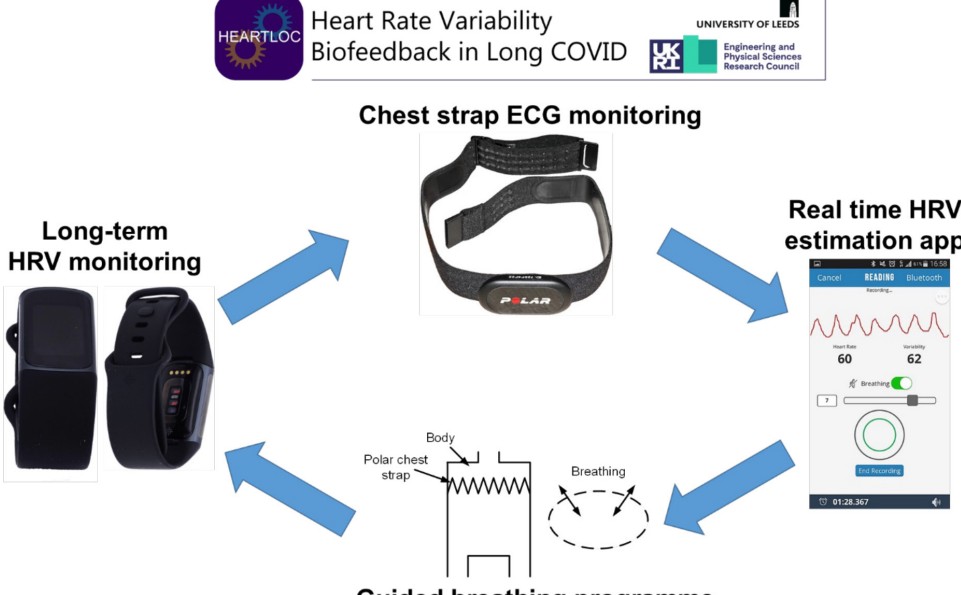

**Figure 1** Heart rate variability biofeedback (HRV-B) using a breathing technique and chest strap for real-time HRV monitoring. Polar H10 picture from Wikimedia commons, reprinted under CC BY-SA 3.0 license. EliteHRV screenshot from Wikimedia commons, reprinted under CC BY-SA 4.0 license.

The exclusion criteria are

► Unable to use the wearable or smartphone app technology.
► Cognitive difficulties or mental health disorders causing inability to consent.
► Any cardiac arrhythmias that are being planned for further investigations and specialist management in the Cardiology service.
► Any unstable cardiorespiratory disease that needs further medical interventions (except asthma management).

### Equipment and technology

To collect medium-term HRV data, participants will wear a Fitbit Charge 5 smartwatch for a total of 6 weeks. The HRV-B itself will be conducted using a Polar H10 chest strap for 10 min two times per day. This connects via Bluetooth to the Elite HRV smartphone app, which is downloaded to participants' phones. Participants will aim to increase their HRV score as displayed in Elite HRV in real time using a diaphragmatic breathing technique (figure 1). Omron M2 blood pressure monitor (endorsed by the British Hypertension Society) will be used to conduct NLT in clinic and the adapted Autonomic Profile (aAP).[27]

### Study phases

The study will be carried out in the following three phases:
► Pre-HRV-B phase.
► HRV-B phase.
► Post-HRV-B phase.

### Pre-HRV-B phase

The participant will either be invited to a research clinic or visited at their home by a member of the research team (first appointment A1). They would have already received the participant information sheet (PIS) at screening and would have had more than 24 hours to read and understand the content of the PIS. Written consent will be signed by the participant and the researcher during this first visit. Devices and baseline outcome measures used in this stage are:

► Fitbit charge five device and the Fitbit smartphone application: The participant will be requested to have the Fitbit device on most of the time during the 6-week period. The application records HRV at night along with other measures of sleep (sleep stages, HR) and daytime activity (such as step count).
► C19-YRSm: the COVID-19 Yorkshire Rehabilitation Scale (C19YRS) is the literature's first condition-specific patient recorded outcome measure, which has been validated in the LC population.[28 29] The modified scale provides a symptom severity score (out of 30), functional disability score (out of 15), other symptoms score (out of 25) and overall health score (out of 10).[23] The participant will complete C19YRSm (online supplemental file 1) at weekly intervals to monitor the impact of the intervention on LC symptoms. They will also have weekly telephone reviews with study researchers for troubleshooting and to ensure maximal compliance with the study.
► Composite Autonomic Symptom Score (COMPASS): the COMPASS 31 will be completed by the participant at the initial visit and again 6 weeks later at the end of the study. Autonomic symptoms are scored for different domains including orthostatic intolerance, vasomotor, secretomotor, gastrointestinal, bladder and pupillomotor. Total scores for each domain are multiplied by a set weighting and then added together

to provide a score out of 100 representing severity of autonomic symptoms. A higher score represents greater severity.[30]

► aAP: this is an autonomic profile test developed by St Mary's Hospital and the National Hospital for Neurology and Neurosurgery and later adapted for domiciliary use during the COVID-19 pandemic.[27] Participants are asked to monitor their heart rate and BP on lying, and at 3 min of standing at various intervals over 24 hours, including after waking, after eating breakfast/lunch/dinner, before and after 5 min of exercise, and before bed (online supplemental file 2). Abnormal results are calculated using the same criteria for heart rate and BP differences as the NLT (HR increase >30/min or BP drop >20 mm Hg).

► WHO Disability Assessment Schedule (WHODAS): this is validated generic measure of functioning and disability. The 36-item scale captures six domains of life (cognition, mobility, self-care, getting along, life activities and participation) with a summary score ranging from 0 (no disability) to 100 (full disability)[31 32]

► EQ-5D-5L: The EQ-5D-5L instrument, provided by the EuroQol Group, is one of widely used quality of life measures, consists of five items covering: mobility, self-care, usual activities, pain/discomfort and anxiety/depression.[33] The item scores can be converted into a total index score by applying health preference weights elicited from a general population. This index score can also be used in economic evaluations to assess the cost-effectiveness of health interventions.[34]

The A1 appointment will last approximately 2 hours and may be longer for those with cognitive fatigue or 'brain fog'. If felt necessary, it will be divided into two 1-hour visits to reduce cognitive fatigue.

## HRV-B phase

One week after the A1 appointment, the participant will be either invited to attend a research clinic or visited at home by a researcher (second appointment A2) to commence the HRV-B study phase. This involves:

► Polar H10 chest strap and Elite HRVB application: The participant will be familiarised with the technology and introduced to a paced breathing regimen via a one-to-one demonstration. They will be instructed to perform the breathing technique using the application at least two times a day, 10 min per session, for a period of 4 weeks. The chest strap device will record HRV for the duration of the session, and the data get recorded in the application. While this phase is ongoing, participants will continue to wear the Fitbit Charge 5 device for the duration of this phase.

► Fitbit charge 5 device and the Fitbit smartphone application.

► C19-YRSm.

## Post-HRV-B phase

The participant will be asked to stop the HRV-B intervention after completing 4 weeks of the treatment. They will

be asked to continue using the Fitbit device for another week when not doing the intervention. They will then either be invited to a research clinic or be visited at home by a study researcher. At this appointment (A3), the participant will complete:

► C19 YRSm: the C19-YRSm will be completed by the patient every week for a total of 6 weeks. There will be a total of 7 C19-YRS documents completed.

► Fitbit charge 5 device and the Fitbit smartphone application.

► NLT and aAP.

► COMPASS 31.

► WHODAS.

► EQ-5D-5L.

During the A3 appointment, the Polar H10 strap and the Fitbit device will be retrieved.

The participants will be invited to complete a further C19-YRS, by email or postal 4 weeks after completion aAP for 24 hours and to email or post the results to the study researcher.

## Outcome measures

*The primary outcome measure* is the C19YRSm, a self-reported patient-reported outcome measure to assess LC symptom severity, functional disability and overall health status.
*Secondary outcome measures* include:
Heart rate measures from chest strap:

► 7-day average HRV score out of 100—quantified by the Elite HRV app via the root mean square of successive differences between normal heartbeats (rMSSD). A natural log (ln) is applied to this figure and then expanded to generate a 1 to 100 score.

► Mean R–R interval.

► Heart rate.

► rMSSD.

► SDNN (SD of Normal to Normal intervals).

► Total power.

► Low-frequency power (LF).

► High-frequency power (HF).

► LF:HF ratio.

Fitbit data:

► Sleep staging data.

► Resting heart rate.

► Daily activity levels, for example, step count and exercise type and duration.

Patient-reported outcome measures:

► NLT and aAP.

► COMPASS 31.

► WHODAS.

► EQ-5D-5L.

During our final interaction with participants in the study, we will ask them the following questions to assess the feasibility of the study:

1. 'How did you find using the technology?'
2. 'How did you find the breathing intervention?'
3. 'Have you noticed any change in your symptoms?'

Their opinions and suggestions will be recorded as quotes in their participant files. However, we are not planning to

**Table 1**  Outcome measures summary schedule

|  | Initial assessment Clinic | Pre HRV-B phase (1 week) | HRV-B phase (4 weeks) | Post HRV-B phase (1 week) |
|---|---|---|---|---|
| Autonomic screening (NLT) | √ |  |  | √ |
| Autonomic function (COMPASS 31) |  | √ |  | √ |
| Home autonomic test (aAP) | √ |  |  | √ |
| Fitbit wrist strap HRV, sleep data |  | √ | √ daily | √ |
| Polar H10 chest strap HRV data |  |  | √ daily |  |
| LC-specific PROM C19-YRSm |  | √ | √ weekly | √ |
| Daily function (WHODAS) |  | √ |  | √ |
| Quality of life (EQ5D-5L) |  | √ |  | √ |

aAP, adapted Autonomic Profile; COMPASS 31, Composite Autonomic Symptom Score; C19-YRSm, modified COVID-19 Yorkshire Rehabilitation Scale; EQ-5D-5L, EuroQol 5-Dimension 5-Level; HRV-B, heart rate variability biofeedback; LC, long COVID; NLT, NASA Lean Test; WHODAS, WHO Disability Assessment Schedule.

undertake a formal qualitative analysis of responses as it is not one of the main objectives of this study.

A summary of the schedule for the completion of outcome measures is shown in table 1.

## SAMPLE SIZE

A formal sample size calculation is not required for a feasibility study as it does not mimic a definitive randomised trial and aim is not to measure effect size.[35] A sample size of 30 is the average sample size across feasibility studies and is accepted as reasonable size to assess the acceptability and suitability of the intervention.[36]

## STATISTICAL ANALYSIS

Quantitative data from standardised questionnaires will be scored as per standard procedures. Data downloaded from the wearable devices will be extracted, cleaned and summarised using specific software packages, including Matlab and Python. Quantitative data will be analysed with simple descriptive statistics. The presence and magnitude of pre and postintervention differences will be examined using repeated paired sample t tests (with Bonferroni adjustment for multiple comparisons), and the effect size will be explored using both Analysis of Variance (ANOVA) partial $\eta^2$, and Cohen's d. Additional exploratory analyses may also be performed to fully analyse the data set produced, guided by the findings of the descriptive statistics.

## Patient and public involvement

Members of the Patient Advisory Group (PAG) with lived experience of LC have been involved in the design, development and delivery of the project. Members of the PAG attended proposal research planning meetings and shared their experiences on symptoms of dysautonomia, which helped shaped the research question, design and outcome measures of this study. Members of this group have contacts with wider patient community groups and helped disseminate information about the study. The PAG meets quarterly with the research team to review progress, ensure the research continues to answer relevant issues and that findings can inform LC care. The group will be involved in the dissemination of research findings and writing lay summary reports that will be shared with the participants.

## Ethics and dissemination

The study has received ethical approval from Health Research Authority (HRA) Leicester South Research Ethics Committee (21/EM/0271). Informed consent will be obtained from all participants. Potential participants will have a minimum of 24 hours to review the PIS and discuss queries with the researcher prior to signing the written consent. General Data Protection Regulation (GDPR) rules will be strictly followed for all data gathered during the study. All data will be fully anonymised as soon as practical. All devices used are (Conformité Européenne (CE) marked and are being used for their intended purposes. There is potential for minor skin irritation from wearing the Fitbit and Polar H10 devices. This will be enquired about at each weekly telephone review.

For participants with cognitive fatigue or 'brain fog' relating to LC, the length of the appointments with the researcher (A1, A2 and A3) may be longer than normal. Supplementary written information will be provided, and if necessary, each of these appointments may be conducted in two shorter sessions to reduce information overload and possible impact on LC symptoms. Participants will be advised that they do not need to proceed with the appointments or the study if they do not want to. All appointments other than the initial NLT can occur at the participants' homes to reduce travel and inconvenience. Participants are free to withdraw at any point in the study. They will be encouraged to give reasons for the withdrawal, but it will not be compulsory to give a reason for withdrawal.

Dissemination will include both academic publications and lay summaries in various formats. Academic outputs will include both medical and engineering literature. Policy impact will be aided by our strong existing links to NHS England and the UK Long COVID National Task Force. Dr Sivan, who leads the NIHR project Long COVID Multidisciplinary consortium for Optimising Treatments and Services across the NHS (LOCOMOTION),[37] is also an advisor for the WHO—Europe on COVID-19

rehabilitation and is also involved in the WHO working party developing a core set of outcome measures for LC.

**Acknowledgements** The authors would like to thank individuals with autonomic problems in long covid and healthcare professionals from the Leeds Covid Rehabilitation service who provided valuable suggestions and feedback during the iterative process of development of this protocol. We are grateful to the Patient Advisory Group for its involvement in all stages of this study.

**Contributors** MS and AC conceptualised the study. MS, AC and RJO'C were awarded EPSRC IAA pump-priming grant for the feasibility study with MS as the Principal Investigator. JC, SH, NP, DW, RT, JD, ADS, RJO'C, AC and MS contributed to the study design and obtained ethical approval. JC and MS wrote an initial draft of the paper by adapting the grant proposal and the ethics protocol. All authors approved the final manuscript. All authors will contribute to recruitment, data acquisition and analysis of the study findings. MS is the corresponding author and guarantor.

**Funding** This research is supported by IAA EPSRC [Ref 112538] with University of Leeds as the sponsor organisation and the Leeds Community Healthcare NHS Trust Covid Rehabilitation service as the research site organisation.

**Competing interests** Manoj Sivan is an advisor to the WHO for the Long COVID policy in Europe.

**Patient and public involvement** Patients and/or the public were involved in the design, or conduct, or reporting, or dissemination plans of this research. Refer to the Methods section for further details.

**Patient consent for publication** Not applicable.

**Provenance and peer review** Not commissioned; externally peer reviewed.

**Data availability statement** Not applicable as this is the protocol paper with no patient data included.

**ORCID iDs**
Stephen Halpin http://orcid.org/0000-0002-0417-8928
Nick Preston http://orcid.org/0000-0001-8429-7320
Rory J O'Connor http://orcid.org/0000-0002-4643-9794
Alexander Casson http://orcid.org/0000-0003-1408-1190
Manoj Sivan http://orcid.org/0000-0002-0334-2968

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
