## [Reviewer comments · BMJ Open]

ARTICLE DETAILS

TITLE (PROVISIONAL)	HEART rate variability biofeedback for Long Covid symptoms (HEARTLOC): protocol for a feasibility study
AUTHORS	Corrado, Joanna; Halpin, Stephen; Preston, Nick; Whiteside, Diana; Tarrant, Rachel; Davison, Jennifer; Simms, Alexander; O'Connor, Rory J.; Casson, Alexander; Sivan, Manoj

VERSION 1 – REVIEW

REVIEWER	Chaudhary, Ashish Hertfordshire Community NHS Trust
REVIEW RETURNED	02-Sep-2022

GENERAL COMMENTS	A valuable study which aims to further develop on the link between dysautonomia and Covid infection. This may lead to a real world, accessible and available treatment option for large numbers of patients with Long Covid associated dysautonomia.
--

REVIEWER	Makimoto, Hisaki Heinrich-Heine-Universitat Dusseldorf
REVIEW RETURNED	18-Oct-2022

GENERAL COMMENTS	General comments: Dr. Corrado, et al. presented their protocol for a study “HEART rate variability biofeedback for Long Covid symptoms (HEARTLOC)” in this manuscript. This is an uncontrolled open-label study of a home use wearable-based heart rate variability biofeedback (HRV-B) technique. The aims of this study are to test the feasibility and to estimate the efficacy of HRV-B in long covid (LC) patients. They will enroll 30 adult LC patients with symptoms of palpitations or dizziness who should have an abnormal NASA Lean Test. The patients undergo a 4-week HRV-B intervention and observation of HRV using wearable devices (Polar chest strap, Fitbit). Under the circumstances that so many wearable devices with personal health records are available, this study seems interesting, and the motivation for using those devices in clinical studies should be appreciated. It will result in a novel health care structure not only for patients but also for healthy individuals. However, the reviewer has some concerns to be cleared as follows before the manuscript should be published. Major comments: 1. The reviewer understood that the authors will investigate the efficacy of HRV-B intervention on autonomic function (HRV) in patients with LC. Then the authors should test if the manipulation during the study does not affect HRV in this population. As the
---

	authors showed in the supplementary files, the collaboration/communication between the study nurses and the participants is mandatory during fulfilling the documents and performing HRV-B maneuvers. Then these communications may affect the autonomic function and the mental status of the participants. For this purpose, a crossover design can be a candidate method. 2. Do the LC symptoms result from autonomic dysfunction? Is it not possible that both the LC symptoms and the autonomic dysfunction can result from the common cause, and we see only the results? If this is the case, the authors should test the autonomic function as a confounding factor during investigating the direct effect of HRV-B on the LC symptoms. 3. In addition to the first and second points, the authors presented the primary outcome measure as C19YRSm. Then the authors intend to investigate the effect of HRV-B on the LC symptoms. In this case, the mentorship of the study nurses may affect the symptoms and the mental status of the participants (i.e. the mentorship should be a confounder). 5. The second objective, “Are consumer grade monitors appropriate technology to use for HRV-B in the domiciliary setting?”: the definition of “appropriate” to test this objective is not clear in the current manuscript. Minor comments: 1. In the part of “Pre-HRV-B Phase”, the measurements during this phase are not completely listed in the manuscript.
--	--

VERSION 1 – AUTHOR RESPONSE

Reviewer: 1

Comments to the Author:

A valuable study which aims to further develop on the link between dysautonomia and Covid infection. This may lead to a real world, accessible and available treatment option for large numbers of patients with Long Covid associated dysautonomia.

Response: Thank you

Reviewer: 2

Comments to the Author:

General comments:

Dr. Corrado, et al. presented their protocol for a study “HEART rate variability biofeedback for Long Covid symptoms (HEARTLOC)” in this manuscript.

This is an uncontrolled open-label study of a home use wearable-based heart rate variability biofeedback (HRV-B) technique. The aims of this study are to test the feasibility and to estimate the efficacy of HRV-B in long covid (LC) patients.

They will enroll 30 adult LC patients with symptoms of palpitations or dizziness who should have an abnormal NASA Lean Test. The patients undergo a 4-week HRV-B intervention and observation of HRV using wearable devices (Polar chest strap, Fitbit).

Under the circumstances that so many wearable devices with personal health records are available, this study seems interesting, and the motivation for using those devices in clinical studies should be

appreciated. It will result in a novel health care structure not only for patients but also for healthy individuals.

Response: Thank you

However, the reviewer has some concerns to be cleared as follows before the manuscript should be published.

Major comments:

The reviewer understood that the authors will investigate the efficacy of HRV-B intervention on autonomic function (HRV) in patients with LC. Then the authors should test if the manipulation during the study does not affect HRV in this population. As the authors showed in the supplementary files, the collaboration/communication between the study nurses and the participants is mandatory during fulfilling the documents and performing HRV-B manoeuvres. Then these communications may affect the autonomic function and the mental status of the participants. For this purpose, a crossover design can be a candidate method.

Response: Thanks for the suggestion. This is a feasibility study to investigate whether this intervention can be tried in this condition rather than estimate the effect size (hence cross over design is not needed at this stage but we will consider this design for future studies). Regarding measuring change in HRV and autonomic function, we are capturing the entire range of HRV measures (see page 10) from the chest strap and Fitbit.

Do the LC symptoms result from autonomic dysfunction? Is it not possible that both the LC symptoms and the autonomic dysfunction can result from the common cause, and we see only the results? If this is the case, the authors should test the autonomic function as a confounding factor during investigating the direct effect of HRV-B on the LC symptoms.

Response: We would like to highlight that dysautonomia is one of the major physiological mechanisms driving long covid symptoms and separating the two would not be scientifically sound. In this study, we are therefore trying to modulate the autonomic system to improve long covid symptoms.

In addition to the first and second points, the authors presented the primary outcome measure as C19YRSm. Then the authors intend to investigate the effect of HRV-B on the LC symptoms. In this case, the mentorship of the study nurses may affect the symptoms and the mental status of the participants (i.e., the mentorship should be a confounder).

Response: The reason for choosing C19-YRSm as primary outcome measure is that its more meaningful for patients and we can infer clinical significance using MCID (minimal clinical important difference) values. We will also have the HRV measures (secondary measures) to compare the pre and post values and correlate this to the C19-YRS changes. The mentorship of the investigator is a potential confounder, but we are not aiming to understand the real effect size in this feasibility study.

The second objective, "Are consumer grade monitors appropriate technology to use for HRV-B in the domiciliary setting?": the definition of "appropriate" to test this objective is not clear in the current manuscript.

Response: We will asking participants whether they were able to use all the equipment at home independently and whether the intervention is acceptable to them (page 10 highlighted section)

Minor comments:

In the part of "Pre-HRV-B Phase", the measurements during this phase are not completely listed in the manuscript.

Response: Thanks for pointing this out. We have now moved the relevant sections of the manuscript to Pre-HRV-B (page 8)

VERSION 2 – REVIEW

REVIEWER	Makimoto, Hisaki Heinrich-Heine-Universität Dusseldorf
REVIEW RETURNED	02-Nov-2022
GENERAL COMMENTS	The authors responded adequately to my concerns. Thank you. The reviewer has no more questions.